# Phosphorous Magnetic Resonance Spectroscopy to Detect Regional Differences of Energy and Membrane Metabolism in Naïve Glioblastoma Multiforme

**DOI:** 10.3390/cancers13112598

**Published:** 2021-05-26

**Authors:** Lisa Maria Walchhofer, Ruth Steiger, Andreas Rietzler, Johannes Kerschbaumer, Christian Franz Freyschlag, Günther Stockhammer, Elke Ruth Gizewski, Astrid Ellen Grams

**Affiliations:** 1Department of Neuroradiology, Medical University of Innsbruck, 6020 Innsbruck, Austria; Lisa-Maria.Walchhofer@tirol-kliniken.at (L.M.W.); Ruth.Steiger@i-med.ac.at (R.S.); Elke.Gizewski@i-med.ac.at (E.R.G.); Astrid.Grams@i-med.ac.at (A.E.G.); 2Neuroimaging Research Core Facility, Medical University of Innsbruck, 6020 Innsbruck, Austria; 3Department of Neurosurgery, Medical University of Innsbruck, 6020 Innsbruck, Austria; J.Kerschbaumer@i-med.ac.at (J.K.); Christian.Freyschlag@i-med.ac.at (C.F.F.); 4Department of Neurology, Medical University of Innsbruck, 6020 Innsbruck, Austria; Guenther.Stockhammer@i-med.ac.at

**Keywords:** MRS, magnetic resonance spectroscopy, energy metabolism, glioblastoma, ATP, PCr

## Abstract

**Simple Summary:**

Glioblastoma multiforme is a highly aggressive brain tumor, tending to infiltrate even larger zones of brain tissue than visible on conventional magnetic resonance imaging. By application of phosphorus magnetic resonance spectroscopy in patients with naïve glioblastoma multiforme, we tried to demonstrate changes in energy and membrane metabolism not only in affected regions but also in distant brain regions, the opposite brain hemisphere, and in comparison to healthy volunteers. We found reduced energetic states and signs of increased cell membrane turnover in regions of visible tumor and differences to and between the “normal-appearing” brains of glioblastoma patients and the brains of healthy volunteers. Our pilot study confirmed the feasibility of the method, so differences between various genetic mutations or clinical applicability for follow-up monitoring can be assessed in larger cohorts.

**Abstract:**

*Background*: Glioblastoma multiforme (GBM) is a highly malignant primary brain tumor with infiltration of, on conventional imaging, normal-appearing brain parenchyma. Phosphorus magnetic resonance spectroscopy (31P-MRS) enables the investigation of different energy and membrane metabolites. The aim of this study is to investigate regional differences of 31P-metabolites in GBM brains. *Methods*: In this study, we investigated 32 patients (13 female and 19 male; mean age 63 years) with naïve GBM using 31P-MRS and conventional MRI. Contrast-enhancing (CE), T2-hyperintense, adjacent and distant ipsilateral areas of the contralateral brain and the brains of age- and gender-matched healthy volunteers were assessed. Moreover, the 31P-MRS results were correlated with quantitative diffusion parameters. *Results*: Several metabolite ratios between the energy-dependent metabolites and/or the membrane metabolites differed significantly between the CE areas, the T2-hyperintense areas, the more distant areas, and even the brains of healthy volunteers. pH values and Mg^2+^ concentrations were highest in visible tumor areas and decreased with distance from them. These results are in accordance with the literature and correlated with quantitative diffusion parameters. *Conclusions*: This pilot study shows that 31P-MRS is feasible to show regional differences of energy and membrane metabolism in brains with naïve GBM, particularly between the different “normal-appearing” regions and between the contralateral hemisphere and healthy controls. Differences between various genetic mutations or clinical applicability for follow-up monitoring have to be assessed in a larger cohort.

## 1. Introduction

Glioblastoma multiforme (GBM) is an aggressive primary brain tumor [1,2]. The recommended MRI protocol for standardized diagnosis includes T2- and T1-weighted sequences, with and without the administration of a gadolinium-based contrast agent [3]. With additional sequences, such as diffusion-weighted imaging (DWI), perfusion-weighted imaging (PWI), or proton-based (1H) magnetic resonance spectroscopy (MRS), supplemental information on cellularity, vascularization, and proliferation can be achieved [4,5,6]. A multimodal combination of these sequences reveals an increase of sensitivity and specificity for the diagnosis and interpretation of follow-up imaging [7]. In several earlier studies, it has been shown that metabolic changes can be found not only within the contrast-enhancing (CE) tumor areas [8,9] but also in the peritumoral T2-hyperintense regions or even normal-appearing brains [10,11,12].

On the one hand, phosphorous MRS (31P-MRS) enables us to analyze metabolites of energy metabolism, namely, inorganic phosphate (Pi), phosphocreatine (PCr), and adenosine triphosphate (ATP). On the other hand, phosphomonoesters (PMEs) and phosphodiesters (PDEs), which are closely linked to membrane turnover, are also essayed [13,14,15,16]. Additionally, it is possible to assess free Mg²^+^ by analyzing the chemical shift of β-ATP, which is dependent on the fraction of total ATP complexed to Mg²^+^ ions. pH can be determined from the signal position of Pi [17,18].

The received opinion is that direct quantification of metabolite concentrations is impractical as it is highly dependent on numerous factors like coil sensitivity, field inhomogeneity, and relaxation time. Therefore, it is common practice to apply metabolite concentration ratios instead of absolute values for further investigation [19]. 

So far, there have only been a few studies that have investigated newly diagnosed GBM with 31P-MRS. In these studies, differences in energy metabolite and/or membrane metabolite ratios and differences in pH or Mg^2+^ values between GBM and the contralateral brain or the brains of healthy volunteers have been described [20,21,22,23,24,25]. 

Therefore, the aim of the present study is to investigate regional differences of 31P-MRS metabolites in patients with naïve GBM and to correlate this data with quantitative diffusion parameters that represent an approved method to detect tumor infiltration.

## 2. Materials and Methods

### 2.1. Subjects

Thirty-two (13 female, 19 male) eligible patients, with a mean age of 63 years (between 32 to 79 years) and naïve, later histologically verified GBM, were included in this prospective study. Additional inclusion criteria were age >18 years, eligibility for an MRI examination, and written informed consent prior to the examination. Exclusion criteria were contraindications for MRI examinations and previously treated tumors. The study has been approved by the local ethics committee (AN 5100 325/4.19).

Tumor locations were temporal (N = 10), frontal (N = 9), parietal (N = 4), temporo-occipital (N = 3), temporo-parietal (N = 2), fronto-temporal (N = 2), occipital (N = 1) and thalamic (N = 1). More tumors were found in the right hemisphere (N = 17) than in the left hemisphere (N = 11). Four tumors crossed the midline and involved both hemispheres, with three of them predominantly on the right side.

Molecular pathology results revealed that in 15 patients (46.8%) the O6-alkylguanine DNA alkyltransferase (MGMT) promoter region was methylated and only two patients (6.25%) had an isocitrate dehydrogenase (IDH) mutant. In two cases, the IDH status was not available. In 30 of the cases (93.7%) p53 was mutated; in 24 of the cases (75%), an epidermal growth factor receptor (EGFR) mutation was detectable. In one patient, both parameters were not available. 

The data from healthy brains was derived from a cohort of age- and gender-matched healthy volunteers; this data contains the values from both supratentorial hemispheres.

### 2.2. Magnetic Resonance Spectroscopy

31P-MRS was performed on a 3T whole-body system (Verio, Siemens Medical AG, Erlangen, Germany) with a double-tuned 1H/31P volume head coil (Rapid Biomedical, Würzburg, Germany). For planning of the 31P-MRS, a sagittal-oriented T2-weighted 3D sequence (space) with isotropic resolution and a voxel size of 1.2 × 1.2 × 1.2 mm³ (repetition time (TR) = 3000 ms, echo time (TE) = 412.0 ms, acquisition time (TA) = 2:50) was acquired. The 3D MRS block was acquired, covering the entire cerebrum of the patients. As much volume of the brain as possible was covered by using the following planning protocol: aligning the coronal inclination on the dorsal line of the brain stem and axial arrangement in the subcranial layers (Figure 1). Inclusion of cavities and skull was avoided as fat, bone, and boundary layers would be intruding the 31P spectra.

The volume of interest was recorded with a 8 × 8 × 8 matrix and a field of view (FOV) of 240 × 240 × 200 mm³, resulting in a 30 × 30 mm² voxel size and a slice thickness of 25 mm. Prior to Fourier transformation, the matrix was interpolated to 16 × 16 × 8 mm³, hence yielding a plane grid size of 15 × 15 mm². The sequence was performed as described in a prior study [26,27,28], with a WALTZ 4 proton decoupling, a TR of 2000 ms, a TE (delay) of 2.3 ms, a flip angle (FA) of 60°, and 10 acquisitions for averaging. 

Additionally, every subject underwent MRI of the entire brain for structural imaging with a 12-channel head coil (Siemens Medical AG, Erlangen, Germany): 

Transverse-oriented 3D T1-weighted MPRAGE with a voxel size of 0.9 × 0.7 × 1.2 mm³ (TR = 1750 ms, TE = 3.3 ms, FA = 9°, FOV = 220 mm³, FOV phase = 71.9%, TA = 4:26 min), before and after contrast agent injection, respectively.

Transverse-oriented T2-weighted inversion recovery turbo spin echo sequence with a voxel size of 0.9 × 0.7 × 3.0 mm³ (TR = 7060 ms, TE = 97.0 ms, FA = 150°, FOV = 220 × 187 mm², matrix = 320 × 80%, number of slices = 49, TA = 5:12 min).

Transverse-oriented diffusion-weighted imaging (DWI) sequence, with b-factors = 0 (one single reference image) and 1000 s/mm^2^ with 20 diffusion directions, and a voxel size of 1.8 × 1.8 × 3.0 mm³ (TR = 7500 ms, TE = 95 ms, FOV = 230 × 230 mm², matrix = 128 × 100%, number of slices = 45, TA = 3:02 min).

### 2.3. Data Processing and Analysis

31P-MRS data were exported from the scanner in .rda format (Siemens Medical AG, Erlangen, Germany) and subsequently processed offline in the time domain with the software jMRUI (Verison 5.0 available at http://www.mrui.uab.es (accessed on 1 March 2014)), utilizing the nonlinear least square fitting algorithm AMARES [29]. The fitting model consisted of 15 Lorentzian-shaped exponentially decaying sinusoids, representing the following metabolites, from left to right: phosphocholine and phosphoethanolamine (summed up as PME); Pi, glycerophosphocholine, and glycerophosphoethanolamine (summed up as PDE); macromolecules (MM), PCr, NAD(H), and ATP. The latter consists of two doublets displaying γ-ATP and α-ATP as well as one triplet displaying β-ATP, which have been summarized and divided by three (Figure 2).

The following brain areas were analyzed with one or more 31P-MRS voxels in each patient: CE tumor areas, not CE T2-hyperintense areas, normal-appearing brain areas adjacent to the T2-hyperintense areas, normal-appearing brain areas more distant on the ipsilateral hemisphere, and normal-appearing brain tissue areas on the contralateral hemisphere (Figure 1). 

The number of analyzed voxels per area of interest was in direct relationship to the size of the respective area, especially of the tumor, and the quality of the spectra. Quality analysis of the spectra was performed according to the criteria of existing literature [30]. Voxels that did not fulfill the quality criteria were discarded from further analysis. Every single spectrum was visually assessed and subsequently matched by an experienced neuroradiologist (A.R.), considering T2-weighted and CE T1-weighted sequences, in order to choose the areas of interest. Considering the large voxel size and the “voxel bleeding” effect due to point spread function (PSF), contamination with other tissues like necrotic areas, cerebrospinal fluid, and bone could not be excluded completely. PSF is a well-known major limitation of MRS. In order to avoid these contaminations and suppress unwanted side lobes of a sinc-shaped PSF, our MRS sequence [31] was acquired with Hamming weighting. After Fourier transformation the nominal cubic voxel yielded a spherical-shaped area of interest increased by about 190%. However, due to the missing side lobes, one can assume that the majority of our signal originates from the effective voxel size, which corresponds with our desired tissue [32]. We performed a simulation on PSF to quantify this effect. A description of the simulation can be found in the Appendix A. Additionally, we tried to overcome possible contamination of undesired tissue by invariably choosing voxels with more than two-thirds of the desired tissue involvement, which led to the exclusion of 20% of the total voxel number. Similarly, if necrotic areas took up to more than one-third of a voxel, the voxel was excluded from further analysis. From the areas under the curve of the different metabolites, the metabolite ratios PCr/ATP, Pi/ATP, PCr/Pi, PME/PDE, PME/ATP, PME/Pi, PME/PCr, PDE/ATP, PDE/Pi, and PDE/PCr have been calculated for the different areas of interest. For the ATP-containing ratios, all three ATP sinusoids were summed up and then divided by three. In addition, pH and Mg^2+^ were extracted from spectroscopic data using the formula published by Iotti et al. [17,18].

In order to detect possible connections between the present 31P-MRS results with a more established method, DWI was investigated. Therefore, regions of interest were manually drawn in the department’s clinical picture archiving and communication system (AGFA, Mortsel, Belgium) on ADC maps and automatically copied to b1000 and FA maps, which is a well-established method [33]. Mean ADC values were given as 10^−3^ mm²/s, b1000 values as s/mm², and fractional anisotropy (FA) as unitless. 

This was possible in 31 patients, as in one patient, no DWI sequence was available pre-therapy. Measurements were performed in the CE areas, the T2-hyperintense areas, and those adjacent to the latter, as, for these regions, data from other studies exist [12,34].

Descriptive statistics and further analyses were investigated with the software GraphPad Prism (Prism 8, GraphPad Software Inc., San Diego, CA, USA). Statistical evaluation was performed with the values of each voxel. As a first step, outliers were identified using the ROUT method and excluded from statistical analysis. Normal distribution of metabolite ratios was assessed with the one-sample Kolmogorov–Smirnov test, applying a significance level of 5%. As the data was not normally distributed, group differences and multiple comparisons were investigated with the Kruskal–Wallis test. Correlation analyses were evaluated with Spearman’s rank correlation coefficient.

## 3. Results

Mean values and standard deviations of the different metabolite ratios and the pH and Mg^2+^ values in the regions of interest are given in Table 1 and visualized in Figure 3.

The PCr/ATP ratio was significantly lower in CE areas in comparison to all the other areas and the brains of healthy volunteers. The highest values were found in the areas adjacent to the T2-hyperintense areas (Figure 3a). The PCr/Pi ratio was significantly lower in CE areas in comparison to all other areas and the brains of healthy volunteers. The highest values were found in the distant ipsilateral areas (Figure 3b). Similarly, also for the Pi/ATP ratio, significantly higher values were found in the CE areas in comparison to all other areas and the brains of healthy volunteers. The lowest values were found in the distant ipsilateral areas (Figure 3c). For the PME/PDE ratios, significantly higher values were found in the CE areas in comparison to all other areas and the brains of healthy volunteers. The lowest values were found in the contralateral brain and healthy volunteers (Figure 3d).

The PME/ATP, PME/Pi, and PME/PCr ratios showed significantly different values between some areas, mainly between the CE areas and all other areas (Figure 3e–g). The PDE/ATP and PDE/PCr ratios showed no difference between the CE and T2-hyperintense areas and significantly increasing values to the other areas (Figure 3h,j). The PDE/Pi ratio showed the lowest values in the CE areas and a significant increase to the distant ipsilateral areas, with no differences between distant ipsilateral, contralateral, or the brains of healthy volunteers (Figure 2i).

The pH values showed no difference between the CE and T2-hyperintense areas and decreasing values to the more distant areas, with no differences between distant ipsilateral, contralateral, or the brains of healthy volunteers (Figure 3k). For the Mg^2+^ concentration, significant differences were found between some of the areas, amongst others, between the CE areas and healthy volunteers (Figure 3l).

Significantly lower ADC values were found in CE areas and the adjacent areas in comparison to T2-hyperintense areas (*p* < 0.0001, Figure 4a). Significantly higher b1000 values were found in the CE and T2-hyperintense areas compared to the adjacent areas (Figure 4b). FA values were significantly higher in the adjacent areas compared to CE and T2-hyperintense regions (Figure 4c). 

In the CE areas, moderate negative correlations were found between the ADC values and the PCr/ATP (*p* = 0.003; r = −0.45) and PME/PDE ratios (*p* = 0.045; r = −0.33). A weak positive correlation was found between the FA values and the PCr/ATP ratios (*p* = 0.017; r = 0.37). A weak positive correlation between the b1000 values and the PME/PDE ratios (*p* = 0.044; r = 0.33) and a moderate positive correlation between the b1000 and pH values (*p* = 0.01; r = 0.46) were also found. In the T2-hyperintense areas, a weak positive correlation was found between the ADC values and the PME/PDE ratio (*p* = 0.04; r = 0.34). In the areas adjacent to the latter, a weak negative correlation was found between the b1000 values and the PCr/Pi ratio (*p* = 0.04; r = −0.37), and moderate positive correlations were found between the b1000 values and the PCr/ATP ratio (*p* = 0.014; r = 0.44) and between the FA values and the Mg^2+^ values (*p* = 0.01; r = 0.46).

Due to the relatively small sample size and low genetic variability, a subgroup analysis of mutations was not feasible.

## 4. Discussion

The main finding of the present study is that the values of the three energy-related ratios, the membrane-related ratio, and, in some cases, the membrane/energy ratios differ significantly between the CE tumor areas and all other investigated areas, as well as in comparison to healthy volunteers. In some cases, differences between the “healthy-appearing” areas of patients with GBM and between the contralateral hemisphere of GBM patients and healthy volunteers were found. PH values did not differ between the CE and T2-hyperintense areas but were significantly decreased with distance from the CE areas. Mg^2+^ values were higher in CE areas than in the brains of healthy volunteers. 

The present results, in comparison with existing literature, are given in Table 2. Prior studies have compared GBM with either the contralateral hemisphere or healthy controls. The region with higher values is marked with “↑” and the region with lower values with “↓”. If no differences between the regions were found, it was marked with “-”. The table shows that the present results are in accordance with the literature if differences have been found. In one study, no “normal” values were presented, and GBM data was compared to nontumorous pathologies [22].

ATP displays an important source of energy for numerous processes within cells. According to existing literature, PCr can be interpreted as energy storage. Therefore, the PCr/ATP ratio can be used to display changes in the energetic state of cells [36]. The PCr/Pi ratio correlates positively with phosphorylation and also with the oxygenation of tissues [37], so it has been used in former studies to represent oxidative capacity [38,39]. When there is a lack of ATP, the creatine kinase equilibrium will buffer ATP. This results in a decrease of PCr as well as an increase of free creatine and Pi [40]. Hence, the Pi/ATP ratio can be used to display the extent of ATP turnover. The present results, with the highest Pi/ATP values in the CE tumor areas, might underline this interpretation. 

Our results of the lowest PCr/Pi ratios in CE areas agree with Albers et al., where the reverse ratio, the Pi/PCr ratio, was found to be significantly elevated in CE areas [41]. Another study showed higher Pi/PCr and Pi/ATP ratios in GBM tissue in comparison to the healthy-appearing hemisphere, which is consistent with the present data [20]. A further study did not find any differences in the CE areas in GBM for the PCr/ATP or PCr/Pi ratio in comparison to healthy volunteers [23]. In the present study, the lowest values of the PCr/ATP and PCr/Pi ratios, as well as the highest values of Pi/ATP in the CE tumor areas, in comparison to the other areas, could be indicative for reduced tissue oxygenation or energetic state due to the beginning of necrosis but still increased ATP turnover. In the present study, for PCr/Pi and Pi/ATP, significantly higher and significantly lower ratios, respectively, were found in the contralateral hemisphere of GBM patients in comparison to healthy controls.

PME and PDE represent markers for membrane turnover as they are precursors and catabolites of phospholipids. In particular, PME is associated with cell growth, rapid cell membrane synthesis, or membrane turnover [41,42]. The PME/PDE ratio is a surrogate for membrane turnover [43,44]. In one prior study, significantly increased PME/PDE ratios were found in glioma patients compared to healthy volunteers, which is comparable to the present results [21]. Decreasing PME/PDE ratios with increasing distance from the CE tumor might indicate an altered membrane metabolism in the entire GBM brain. 

Generating ratios of the lipid-related metabolites PME and PDE with respect to energy-related metabolites Pi, PCr, and ATP reflects tumor growth and cell reproduction rates [21]. Our findings of higher PME/Pi and PME/PCr and lower PDE/Pi and PDE/PCr ratios endorse the results of previous studies, although, in these studies, CE tumor areas were assessed in comparison to healthy controls without particular attention to regional differences [21,24]. Our study gives hints that these changes do not only affect the CE areas but also the T2-hyperintense areas and the different “normal-appearing” areas and that there are differences to the brains of normal controls.

The present data demonstrate significantly higher pH in CE and T2-hyperintense areas compared to all other areas of interest. These findings follow previous studies [35,45].

Other studies have found significantly higher Mg²^+^ concentrations in tumor regions in comparison to healthy controls, as in the present study [25,46]. We additionally detected higher values in the contralateral hemisphere of the GBM patients in comparison to healthy volunteers.

As mentioned in the introduction, PSF caused by common acquisition techniques in MRS can influence the signals of neighboring voxels. We tried to minimize this effect by using an optimized MRS sequence [31] by thoroughly choosing the tissue of interest and avoiding contamination from other tissue. We found highly significant differences in our regions under investigation in comparison to healthy controls and correlated our findings with well-established diffusion parameters.

The present DWI results, with lower ADC values in CE tumor areas, are partially in accordance with the literature: Wang et al. found decreased ADC values in the CE regions of GBM in comparison to not-CE regions of the tumor. Lower ADC values in the CE areas in comparison to surrounding T2-hyperintense areas were also found in the present study [34]. 

The negative correlation between the PCr/ATP ratio or PME/PDE and the ADC values in the CE areas in the present study may underline the positive connection between energetic state, membrane turnover, and cellularity, as has been described before [19,39].

Another group found significant differences in the ADC results between GBM, peritumoral tissue, and healthy-appearing tissue [11,12]. These are in accordance with our ADC results. Moreover, decreased FA values were described to represent a disruption of normal white matter trajectories due to tumor growth [11]. In the present study, a positive correlation was found between FA values and the PCr/ATP ratio in the CE areas, which can be interpreted as an expression of high energy demand in these areas. In a meta-analysis of coregistered imaging regions of interest and biopsy targets, an inverse correlation of ADC values and cellularity in GBM could be demonstrated, presumably due to the narrowing of extracellular space and restricted water diffusion in the hypercellularity of neoplastic processes [47]. 

In order to find tumor-rich biopsy targets, some studies have tried to characterize histologic heterogeneities in GBM by multimodal MRI, including contrast-enhanced MRI, PWI, DWI [9], or a combination of PWI, 1H-MRS, and 18-fluorethyltyrosin-PET [48]. 31P-MRS might add valuable information to that. In another study, it was possible to detect tumor infiltration and predict future recurrence in patients with GBM from multimodal MRI (T2- and T1-weighted contrast-enhanced imaging, PWI, DWI) [49]. 

There are several limitations of the present study that need to be addressed. First of all, the number of included patients was small. Therefore, further subgroup analyses with regard to genetic mutations were not reasonable. Furthermore, due to the large voxel size, contamination with other tissues like cerebrospinal fluid or bone could not be excluded completely. However, we tried to overcome possible contaminations by choosing voxels with more than two-thirds of the tissue of interest and correlated our findings with well-established diffusion parameters. Moreover, our findings are in excellent accordance with existing literature. Our time-domain model function was composed of 15 exponentially decaying sinusoids in the frequency domain. Initially, two metabolites were included as prior knowledge at a fixed chemical shift of 2.2 ppm, which could be used to account for potential macromolecule signals (MMs) in the phosphodiester region and NAD(H) at −8.3 ppm, if PCr was adjusted to 0 ppm. However, we have to note that due to poor signal-to-noise ratios and, therefore, a noisier baseline of the spectra in the tumor regions, we decided to exclude these two metabolites from our analyses. 

As there are no prior studies that have investigated regional differences of 31P-MRS metabolites in GBM brains, the present study can be considered a pilot study for the feasibility of this method.

## 5. Conclusions

This pilot study shows the feasibility and applicability of 31P-MRS in patients with naïve GBM in a clinical setting. Several regional differences of energy and membrane metabolite ratios, pH, and Mg^2+^ in different regions of patients with naïve GBM have been found. Not only differences between the visible tumor areas and the “normal-appearing” brain areas of GBM patients, but also differences between the contralateral hemisphere of GBM patients and the brains of healthy controls could be detected. This has been described for the first time for this method. This knowledge leads the way to further studies in larger cohorts: dividing into subgroups with different mutations, monitoring patients under therapy, or investigating patients with low-grade gliomas. Furthermore, it could lead to an individualized adaption of radiation therapy or more aggressive surgery.

## Figures and Tables

**Figure 1 cancers-13-02598-f001:**
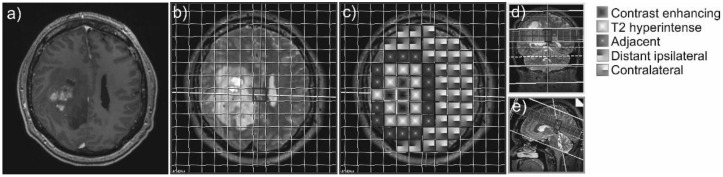
Example of a contrast-enhancing GBM (**a**) and the 31P-MRS measure grid coregistered to the axial T2-weighted sequence of a patient with GBM (**b**). The different areas of interest are coded with different patterns (**c**). On the right side, coronal (**d**) and sagittal (**e**) slices display the position of the voxel slice within the brain.

**Figure 2 cancers-13-02598-f002:**
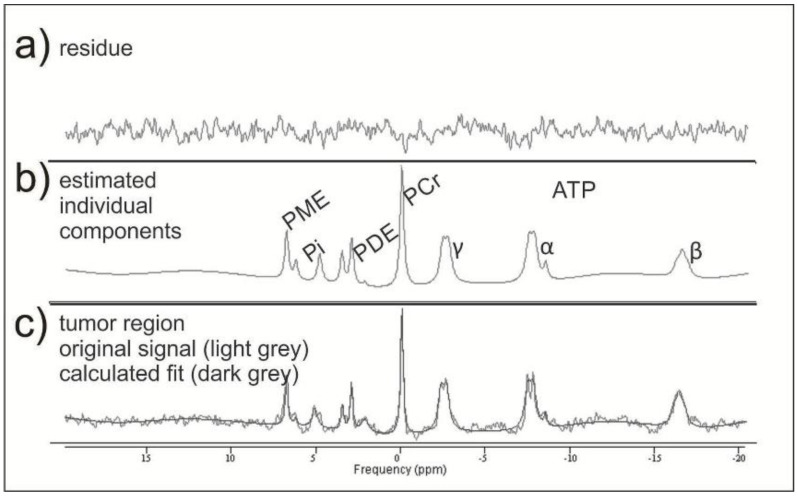
Example of two mathematically fitted 31P-MRS tumor spectra. In the middle row (**b**), there are the estimated individual metabolites with the corresponding residue (**a**) resulting from the mathematical fit. In the lower row (**c**), they are both together.

**Figure 3 cancers-13-02598-f003:**
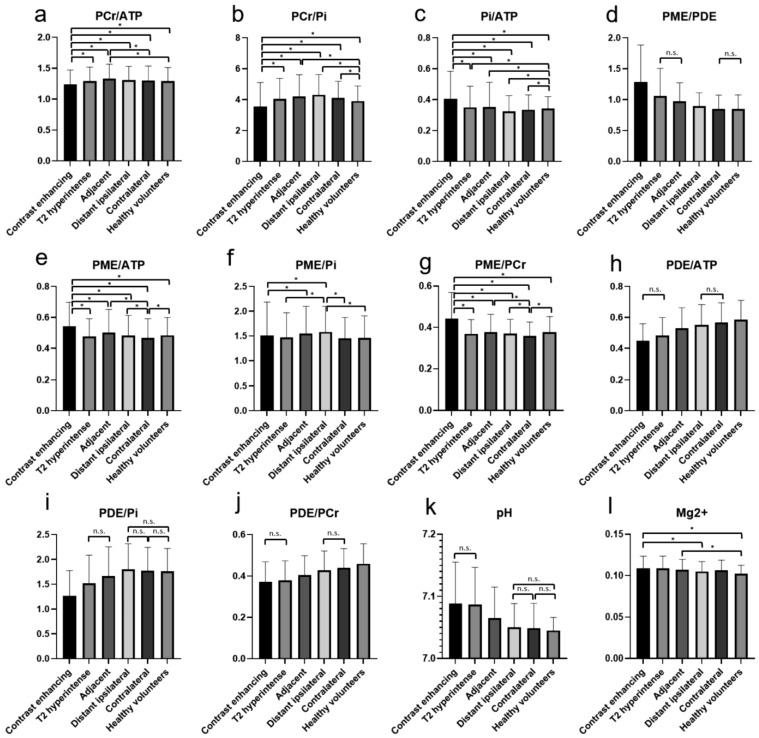
The different metabolite ratios of PCr/ATP (**a**), PCr/Pi (**b**), Pi/ATP (**c**), PME/PDE (**d**), PME/ATP (**e**), PME/Pi (**f**), PME/PCr (**g**), PDE/ATP (**h**), PDE/Pi (**i**), PDE/PCr (**j**), as well as pH (**k**) and Mg^2+^ (**l**), with mean values and standard deviations for the different investigated areas. Significant differences are marked with “*”. For some ratios, in which most differences are significant, the not-significant relationships are marked with a horizontal line and “n.s.”. The values of the *y*-axis (**a**–**j**) are ratios and, therefore, displayed without units.

**Figure 4 cancers-13-02598-f004:**
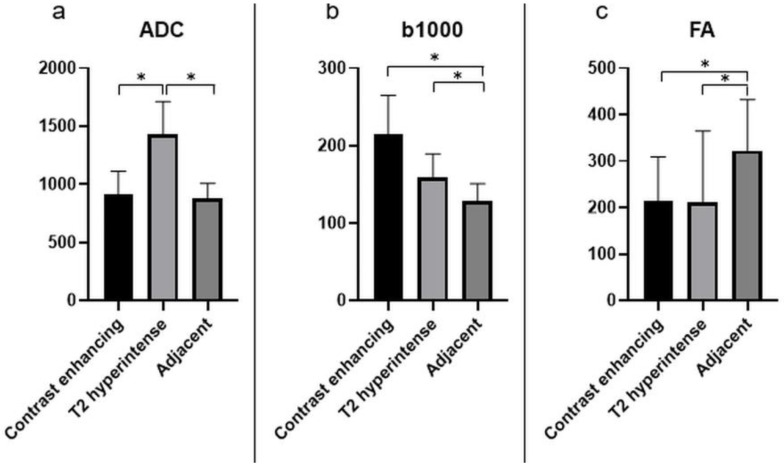
The different quantitative DWI parameters, ADC (**a**), b1000 (**b**), and FA (**c**), with mean values and standard deviations for the relevant investigated areas. The units of the *y*-axis are 10^−3^ mm²/s for ADC, s/mm² for b1000, and unitless for FA. Significant differences are marked with “*”.

**Table 1 cancers-13-02598-t001:** Mean values of standard deviations of metabolite ratios and pH and Mg^2+^ values in mMol.

	Contrast Enhancing	T2 Hyperintense	Adjacent	Distant Ipsilateral	Contralateral	Healthy Volunteers
PCr/ATP	1.24 ± 0.24	1.30 ± 0.22	1.32 ± 0.24	1.30 ± 0.22	1.30 ± 0.23	1.29 ± 0.22
PCr/Pi	3.56 ± 1.56	4.09 ± 1.33	4.21 ± 1.43	4.32 ± 1.31	4.11 ± 1.08	3.91 ± 0.97
Pi/ATP	0.41 ± 0.18	0.35 ± 0.14	0.35 ± 0.16	0.32 ± 0.10	0.33 ± 0.10	0.34 ± 0.08
PME/PDE	1.29 ± 0.60	1.06 ± 0.46	0.97 ± 0.30	0.90 ± 0.21	0.85 ± 0.23	0.85 ± 0.23
PME/ATP	0.54 ± 0.16	0.48 ± 0.11	0.50 ± 0.15	0.48 ± 0.13	0.47 ± 0.13	0.49 ± 0.12
PME/Pi	1.50 ± 0.68	1.47 ± 0.49	1.55 ± 0.55	1.58 ± 0.52	1.45 ± 0.42	1.46 ± 0.44
PME/PCr	0.44 ± 0.13	0.37 ± 0.07	0.38 ± 0.09	0.37 ± 0.07	0.36 ± 0.07	0.38 ± 0.08
PDE/ATP	0.45 ± 0.11	0.48 ± 0.12	0.53 ± 0.13	0.55 ± 0.13	0.57 ± 0.13	0.59 ± 0.13
PDE/Pi	1.27 ± 0.51	1.52 ± 0.57	1.67 ± 0.59	1.80 ± 0.52	1.77 ± 0.47	1.76 ± 0.50
PDE/PCr	0.37 ± 0.10	0.38 ± 0.09	0.41 ± 0.09	0.43 ± 0.09	0.44 ± 0.09	0.46 ± 0.10
pH	7.092 ± 0.07	7.089 ± 0.06	7.066 ± 0.05	7.050 ± 0.04	7.049 ± 0.04	7.045 ± 0.02
Mg^2+^	0.109 ± 0.01	0.103 ± 0.03	0.105 ± 0.02	0.103 ± 0.02	0.106 ± 0.02	0.103 ± 0.01

**Table 2 cancers-13-02598-t002:** Comparison of the present data with data from the literature. The region with higher values is marked with “↑” and the region with lower values with “↓”. If no differences between the regions were found, it was marked with “-”.

	Location	Present Data	Maintz et al. (2002) [23]	Bulakbasi et al. (2007) [25]	Hattingen et al. (2011) [20]	Ha et al. (2013) [21]	Kamble et al. (2014) [22]	Wenger et al. (2017) [35]	Hnilicova et al. (2020) [24]
PCr/ATP	Tumor	↓	-	↓		-	↓		↓
	Control	↑	-	↑		-			↑
PCr/Pi	Tumor	↓	↓	↓	↓	-	↓		↓
	Control	↑	↑	↑	↑	-			↑
Pi/ATP	Tumor	↓	-	-	↑	-	-		↑
	Control	↓	-	-	↓	-			↓
PME/PDE	Tumor	↑				↑	↑		↑
	Control	↓				↓			↓
PME/ATP	Tumor	↑	-	↑		-			↑
	Control	↓	-	↓		-			↓
PME/PCr	Tumor	↑				↑			↑
	Control	↓				↓			↓
PME/Pi	Tumor	-				-			↑
	Control	-				-			↓
PDE/ATP	Tumor	↓	↓	↓		-	-		↓
	Control	↑	↑	↑		-			↑
PDE/PCr	Tumor	↓				-			↓
	Control	↑				-			↑
PDE/Pi	Tumor	↓				-			↓
	Control	↑				-			↑
pH	Tumor	↑	↑	↑		-		↑	↑
	Control	↓	↓	↓		-		↓	↓
Mg^2+^	Tumor	↑		↑					
	Control	↓		↓					

## Data Availability

The data presented in this study are available in Appendix A.

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
