# Peer review of "Phosphorous Magnetic Resonance Spectroscopy to Detect Regional Differences of Energy and Membrane Metabolism in Naïve Glioblastoma Multiforme"

_cancers, 2021, doi:10.3390/cancers13112598_

Round 1
Reviewer 1 Report
Also nowadays rarely used 31P MR spectroscopy promises interesting non-invasive insights in metabolism in normal as well as in pathologic tissues. With the application in cerebral gliomas an interesting application is picked. The study is well designed and the results are conclusive. I have only very few minor remarks:
- You should choose only one spelling for naive/naïve.
- In the materials and methods section the positioning of the MRS is confusing.
Author Response
Also nowadays rarely used 31P MR spectroscopy promises interesting non-invasive insights in metabolism in normal as well as in pathologic tissues. With the application in cerebral gliomas an interesting application is picked. The study is well designed and the results are conclusive. I have only very few minor remarks:
- You should choose only one spelling for naive/naïve.
ANSWER: The spelling for this word has been changed to naïve throughout the manuscript. (page 1 lines 4, 21, 29; page 3, line 79)
- In the materials and methods section the positioning of the MRS is confusing.
ANSWER: We are not precisely sure what this remark is referring to, but we think it might refer to the mixed-up figure captions of figure 1 and 2. We addressed it by changing the figure captions for figure 1 and figure 2 to their correct positions and giving them new positions in the manuscript. (page 4, lines 116 – 120; page 5, lines 154 – 156)
Reviewer 2 Report
General
This paper describes a 31P MRSI study to monitor regional changes in energy and lipid metabolim due to infiltration of glioblastoma. The reviewer feels, that it needs a thorough evaluation of the methodological limits of 31P MRS caused by the poor point spread function (due to k-space sampling of 8x8x8), before addressing diagnostic problems which rely on spatial modifications of metabolite ratios. It also seems to the reviewer, that the final version lacks a rigorous inspection before submission, which is obvious from method section (inconsistent typing of units) and the switching of 2 figure captions.
Details
Abstract
p 1, l 28: “In addition..” Provide the size of the subgroup to indicate the statistical power of the presented results.
p 3, l 104: “TE” This TE is rather a delay caused by the duration of the slice selected pulse. TE may be misleading (although it is stated in the Siemens parameter file) since no echo is recorded.
p 3, l 111: “according…%)” skip, but fix units for FOV (supescript, space before units). Same for later in this paragraph.
p 3, l 116: Whole paragraph before figure; Several point: listing of metabolites from left to right is not correct, Pi should follow PME; two signals which are visible in the fit (NAD(H) and macromolecules at about 1 ppm) are not mentioned in the text. Figure caption matches Figure 2.
p 4, l 136: Figure caption matches Figure 1.
p 4, l 140: “The ….spectra” How does the spectra quality affect the number of voxels. I guess, spectra with bad quality could have been discarded, but there are no criteria mentioned which would lead to discarding
p 4, l 152: “DWI…patients” This reduces the power significantly. Actually, since data were co-registered off-line, DTI data from a different scanner could have been included by co-registering them to the 31P MRSI study
p 4, l 157: Regarding statistic: Was the statistic evaluation performed on voxels or averaged values for each ROI?
p 5, l 180: “The units….readability” These ratios have no units and should not be arbitrary
p 6, l 196: “The main….areas” As long as there is no evaluation of the effect of PSF and a more detailed description of data evaluation (e.g. pooling of voxels from ROI before comparison, paired comparison) the results discussed here are not sufficiently supported. E.g. p 7 l 230: “The decreasing…brain” The finding may just reflect the decrease in voxle bleeding to more distant voxels.
p 7, l 264: “However,….involvment” Unfortunately the authors are not aware about the serious implications of the poor PSF, which is modulating the intensities, even in the center of the FOV, according to the overall geometry of the head.
Author Response
General
This paper describes a 31P MRSI study to monitor regional changes in energy and lipid metabolism due to infiltration of glioblastoma.
- The reviewer feels, that it needs a thorough evaluation of the methodological limits of 31P MRS caused by the poor point spread function (due to k-space sampling of 8x8x8), before addressing diagnostic problems which rely on spatial modifications of metabolite ratios.
ANSWER: The reviewer is correct when pointing out the problematic nature of MR spectroscopy due to limited k-space sampling resulting in poor point spread functions. However, this drawback is an issue which is always present in MRS if not addressed with special sequence programming. To overcome that effect, we utilized a well-established 31P MRS sequence (based on a conventional Siemens sequence) from Dr. Pilatus and Prof. Hattingen [Hattingen E, Lanfermann H, Menon S, Neumann-Haefelin T, de Rochement RD, Stamelou M, Höglinger GU, Magerkurth J, Pilatus U. Combined 1H and 31P MR spectroscopic imaging: impaired energy metabolism in severe carotid stenosis and changes upon treatment. MAGMA. 2009 Feb;22(1):43-52. doi: 10.1007/s10334-008-0148-9. Epub 2008 Oct 15. PMID: 18855032.], as we employed the identical double-tuned 1H/ 31P volume head coil (Rapid Biomedical, Würzburg, Germany). Moreover, we are aware that the actual voxel size is larger than the nominal voxel size and due to residual phase dispersal, the PSF is not uniform across the nominal voxel size leading to only around 87% of the observed signal originating from the targeted spatial location [Maudsley AA Sensitivity in Fourier imaging J Magn Reson 68, 363–366 (1986)]. Additionally, due to the so-called voxel bleeding the remaining signal is spread to adjacent voxels and the exact contribution of the PSF also depends on the points of the spatial grid. These issues of signal contamination may be partially cancelled out in larger objects/regions due to the positive and negative sinc lobes, this is why we pooled our intended voxels from well-demarcated brain regions, and voxels with less than 2/3 of the addressed tissue were completely left out. (page 6, line 175; page 14, lines 181 – 184)
- It also seems to the reviewer, that the final version lacks a rigorous inspection before submission, which is obvious from method section (inconsistent typing of units) and the switching of 2 figure captions.
ANSWER: We apologize for the typographical errors. The manuscript now underwent a thorough check. We changed the figure captions for figure 1 and figure 2 to their correct positions. (page 4, lines 116 – 120; page 5, lines 154 – 156)
Details
Abstract
- p 1, l 28: “In addition..” Provide the size of the subgroup to indicate the statistical power of the presented results.
ANSWER: Initially, we aimed to correlate DTI data with 31P-MRS data using a semi-automatic segmentation tool provided by OLEA. This only worked with some of the datasets from a certain scanner. However, in the meantime we discontinued the use of this software. To sort out this problem, we decided to draw ROIs on the different diffusion series on our PACS system, in which it is possible to copy ROIs from one series to another, and to receive at least signal intensity data of the different regions. This method has demonstrated to be reliable, so we applied it for all patients except for one (this patient did not receive a DWI pre-operatively) and included the data into our statistics. (page 1, line 31; page 6, lines 190 – 199)
- p 3, l 104: “TE” This TE is rather a delay caused by the duration of the slice selected pulse. TE may be misleading (although it is stated in the Siemens parameter file) since no echo is recorded.
ANSWER: We were using the same 31P MRS sequence (slightly adapted regarding the FOV and therefore resulting in a slightly smaller voxel size) after consultation with Dr. Pilatus and as described in Hattingen et al [Hattingen E, Lanfermann H, Menon S, Neumann-Haefelin T, de Rochement RD, Stamelou M, Höglinger GU, Magerkurth J, Pilatus U. Combined 1H and 31P MR spectroscopic imaging: impaired energy metabolism in severe carotid stenosis and changes upon treatment. MAGMA. 2009 Feb;22(1):43-52. doi: 10.1007/s10334-008-0148-9. Epub 2008 Oct 15. PMID: 18855032.], so we referred to TE similarly, but the reviewer is right regarding misleading, hence we adapted the wording to TE(delay) = 2.3ms. (page 4, line 126)
- p 3, l 111: “according…%)” skip, but fix units for FOV (superscript, space before units). Same for later in this paragraph.
ANSWER: We thoroughly checked the entire manuscript with regard to typing of units and corrected them.
- p 3, l 116: Whole paragraph before figure; Several point: listing of metabolites from left to right is not correct, Pi should follow PME;
ANSWER: We adapted the order of the metabolites from left to right. (page 5, line 147)
- Two signals which are visible in the fit (NAD(H) and macromolecules at about 1 ppm) are not mentioned in the text.
ANSWER: The 31P spectra were analyzed in the time domain with jMRUI software tool (Version 5.0, available at http://www.mrui.uab.es) employing the nonlinear least square fitting algorithm AMARES, which is able to handle prior knowledge. Our time domain model function was composed of 15 exponentially decaying sinusoids in the frequency domain. The two metabolites mentioned by the reviewer were included as prior knowledge at a fixed chemical shift of 2.2 ppm which could be used to account for potential macromolecule signals (MM) in the phosphodiester region and NAD(H) at -8.3 ppm (if PCr adjusted at 0 ppm). However, due to the poorer signal to noise ratio, and therefore noisier baseline of the spectra, in tumor regions (compared to healthy brain areas), we decided to exclude these two metabolites from our analyses. (page 5, line 149; page 14, lines 389 – 396)
- Figure caption matches Figure 2.
ANSWER: We changed the figure captions for figure 1 and figure 2 to their correct positions. (page 4, lines 116 – 120; page 5, lines 154 – 156)
- p 4, l 136: Figure caption matches Figure 1.
ANSWER: We changed the figure captions for figure 1 and figure 2 to their correct positions. (page 4, lines 116 – 120; page 5, lines 154 – 156)
- p 4, l 140: “The ….spectra” How does the spectra quality affect the number of voxels. I guess, spectra with bad quality could have been discarded, but there are no criteria mentioned which would lead to discarding
ANSWER: Quality analysis of the spectra was performed according to criteria of existing literature [Kreis R. Issues of spectral quality in clinical 1H-magnetic resonance spectroscopy and a gallery of artifacts. NMR Biomed. 2004 Oct;17(6):361-81. doi: 10.1002/nbm.891. PMID: 15468083]. We added this reference to our manuscript on page 5, lines 169 – 170. Voxels that did not fulfill quality criteria were discarded from further analysis.
- p 4, l 152: “DWI…patients” This reduces the power significantly. Actually, since data were co-registered off-line, DTI data from a different scanner could have been included by co-registering them to the 31P MRSI study
ANSWER: Thank you for that recommendation. In fact, we were also very dissatisfied with that situation. We used the software from OLEA for a certain time in our department, allowing to segment contrast enhancing tumor and T2 hyperintense areas in a semi-automatic manner and to extract the diffusion values after co-registration. This only worked with some of the datasets from a certain scanner. This procedure was intended to serve as parameters to correlate our MRS data with. The idea was, if there are correlations between the diffusion data and the 31P parameters, it could add arguments that our MRS data is reliable. However, after reading your comment we added another approach in drawing ROIs on the different diffusion series on our PACS system, in which it is possible to copy ROIs from one series to another, and to receive at least signal intensity data of the different regions from this. This method has demonstrated to be reliable [El Kady, R.M.; Choudhary, A.K.; Tappouni, R. Accuracy of apparent diffusion coefficient value measurement on PACS workstation: A comparative analysis. AJR. American journal of roentgenology 2011, 196, W280-284, doi:10.2214/ajr.10.4706.]. So we performed this on all patients in the regions “contrast enhancing tumor”, T2-hyperintense areas” and those adjacent to these. For these areas data from other studies are existing, and we can therefore compare our DWI data to the data of prior studies. However, the relations of the different diffusion parameters between the different areas are similar in the initial OLEA measurements and the ROI signal intensity measurements. In addition, the correlations between some of the diffusion values and some MRS ratios are present in the OLEA measurements and the signal intensity measurements. Therefore, we think that our signal intensity values can be used for this purpose and we hope that this is sufficient to answer your request. We changed the manuscript accordingly on page 6 line 190 – 199 and page 9, lines 253 – 258.
- p 4, l 157: Regarding statistic: Was the statistic evaluation performed on voxels or averaged values for each ROI?
ANSWER: The statistical evaluation was performed with the values of each individual voxel. No averaged values of the ROIs were used. The number of used investigated voxels per individual patient was depending on the size of the area to be investigated. We extended this information into the manuscript on page 6, line 202.
- p 5, l 180: “The units….readability” These ratios have no units and should not be arbitrary
ANSWER: We dismissed the term “arbitrary” from the figure caption and replaced it with “displayed without units”. (page 9, lines 251 – 252)
- p 6, l 196: “The main….areas” As long as there is no evaluation of the effect of PSF and a more detailed description of data evaluation (e.g. pooling of voxels from ROI before comparison, paired comparison) the results discussed here are not sufficiently supported. E.g. p 7 l 230: “The decreasing…brain” The finding may just reflect the decrease in voxel bleeding to more distant voxels. p 7, l 264: “However,….involvement” Unfortunately the authors are not aware about the serious implications of the poor PSF, which is modulating the intensities, even in the center of the FOV, according to the overall geometry of the head.
ANSWER: For the different areas of interest, data from individual voxels were used for the statistical evaluation. Depending on the size of the respective areas in the individual patients, the number of investigated voxels per patient were different. That means in one patient it might have been only two voxels for “contrast enhancing tumor”, whereas for another patient it might have been 10 voxels for this area. In addition, the areas “distant ipsilateral” or “contralateral” were much larger, than “contrast enhancing” or “T2-hyperintense” in all patients, therefore, the amount of voxels were different for the different areas of interest. This is why in this study no paired analyses were performed. We had a similar situation in a prior study of our group, in which we performed paired analyses from mean values of different areas of interest. In this mentioned manuscript the reviewers recommended to better investigate data from individual voxels and not from mean values. Since then, the group investigates 31P data this way. We described this in a more detailed way in the materials and methods section on page 6, line 202. We are well aware of the “voxel bleeding” problem from our prior studies and from literature. This might be a problem in many MRS studies and especially 31P MRS studies. However, the present sequence has not been developed recently and it has shown to be reliable. It has been used by other groups and ourselves to different diseases or questions, with reasonable results for the respective research questions. We put a lot of emphasis on the accurate choosing of our voxels (voxel covering mostly the tissue of interest and no other tissues; excluding voxels with bad spectral quality) and the standard deviations were low for all ratios/areas. As depicted in the new “literature table” on page 11 line 302, our analysis generated similar results as described in seven other studies. Regardless of the fact that these studies compared tumor with control, leaving the innumerous ratios and values we investigated. In the present study the standard deviations of the metabolite ratios, pH and Mg values are low, although the tumours were positioned in different supratentorial regions. This might be another hint that the data is reliable and voxel bleed might not be affecting the data in a significant manner. In addition, there are correlations between our data and the more established diffusion values, which make sense. To overcome the problem you mentioned, we performed correlation analyses with diffusion data in the first place. Therefore, we are convinced that our data is reliable and representative for the research question. Nontheless, we put more emphasis on the discussion of PSF and voxel bleeding in a more detailed way on page 13, lines 381 – 386 as potential major limitations.
Reviewer 3 Report
Interesting manuscript comparing 31P energy metabolism in GBM and surrounding human brain and DWI/ADC changes.
The 31PMRS data should be sufficient to provide intracellular pH (PCr to Pi shift) and free Magnesium concentration (chemical shift beta ATP, Iotti et al 1996 amongst others).
The specimen spectra (Fig 1) suggest excellent spectral resolution.
Addition of these data would enhance the manuscript.
At least these techniques could be mentioned in the introduction.
Author Response
Interesting manuscript comparing 31P energy metabolism in GBM and surrounding human brain and DWI/ADC changes.
- The 31PMRS data should be sufficient to provide intracellular pH (PCr to Pi shift) and free Magnesium concentration (chemical shift beta ATP, Iotti et al 1996 amongst others). The specimen spectra (Fig 1) suggest excellent spectral resolution. Addition of these data would enhance the manuscript. At least these techniques could be mentioned in the introduction.
ANSWER: We calculated pH and Mg concentrations for the regions of interest, according to your suggested literature and provided the data in the results section. The results are very interesting, thank you very much for that recommendation! The necessary changes can be found in all chapters throughout the manuscript.
Reviewer 4 Report
The manuscript presents incorret loading of the figures and relative captions that may prevent the analysis of the data. Any statement can be released after reading the correct format.
Novelty: though many 31P MRS studies of phosphometabolites in various brain diseases including glioblastoma have been published, the study on Phosphate metabolites in the entire brain and correlation with MRI identified as different zones is new.
The manuscript reports on differences in metabolism detected by 31P MRS in affected and unaffected brain regions in 32 GBM patients with different tumor locations, before any treatment, in order to correlate the metabolic pattern to MRI performed as quantitative ADC to detect tumor infiltration.
The analysis of phosphorous metabolite ratios is proposed in brain areas classified, by means of MRI, in five types , i.e. contrast enhanced, T2 hyperintense, adjacent, distant, ipsilateral and contralateral . The energy bound metabolites Phosphocreatine, ATP, Inorganic Phosphate and the lipid-related phosphomonoesthers and phosphodiesther are detected and the relative ratios examined.
At present the manuscript needs substantial changes, besides the necessary corrections in figures and captions.
Metabolite ratios should be reported (in tables or in figure captions) for more reliable reading of the corresponding values. ATP values used for the ratios must be checked and assessed (the sum of the three ATP peaks or only γ-ATP)
Lipid related PME and PDE should be reported as ratios at least with respect to ATP (possibly, also to Pi or PCr)
Values fom healthy subjects - either from the authors of the study or from literature data, in the same experimental setting - should be included and agreement with literature data confirmed.
pH values from Pi chemical shift may be included
Minor points
Sentence in lines 146-148 is not clear
Some printing errors: e.g.
Line 84 “promotor” is ”promoter”
Line 124 “displaying-ATP” is “displaying g -ATP”
Check commas and points
Author Response
The manuscript presents incorrect loading of the figures and relative captions that may prevent the analysis of the data. Any statement can be released after reading the correct format. Novelty: though many 31P MRS studies of phosphometabolites in various brain diseases including glioblastoma have been published, the study on Phosphate metabolites in the entire brain and correlation with MRI identified as different zones is new. The manuscript reports on differences in metabolism detected by 31P MRS in affected and unaffected brain regions in 32 GBM patients with different tumor locations, before any treatment, in order to correlate the metabolic pattern to MRI performed as quantitative ADC to detect tumor infiltration. The analysis of phosphorous metabolite ratios is proposed in brain areas classified, by means of MRI, in five types, i.e. contrast enhanced, T2 hyperintense, adjacent, distant, ipsilateral and contralateral. The energy bound metabolites Phosphocreatine, ATP, Inorganic Phosphate and the lipid-related phosphomonoesthers and phosphodiesther are detected and the relative ratios examined.
At present the manuscript needs substantial changes, besides the necessary corrections in figures and captions.
ANSWER: We changed the figure captions for figure 1 and figure 2 to their correct positions. (page 4, lines 116 – 120; page 5, lines 154 – 156)
- Metabolite ratios should be reported (in tables or in figure captions) for more reliable reading of the corresponding values.
ANSWER: Thank you for that recommendation. In the new table 1 all the ratios, including the “new” energy metabolite to membrane metabolite ratios, as well as the pH and the Mg2+ values are given. (page 6, line 213)
- ATP values used for the ratios must be checked and assessed (the sum of the three ATP peaks or only γ-ATP)
ANSWER: All three ATP-integrals were summed up and then divided by three. This has additionally been described in the manuscript on page 5 line 151 and page 6, line 180.
- Lipid related PME and PDE should be reported as ratios at least with respect to ATP (possibly, also to Pi or PCr)
ANSWER: Thank you for that recommendation. We added the ATP/PME, Pi/PME, PCr/PME, ATP/PDE, Pi/PDE, PCr/PDE ratios to table 1, calculated ANOVAs (figure 3). The necessary changes can be found in all chapters throughout the manuscript.
- Values from healthy subjects - either from the authors of the study or from literature data, in the same experimental setting - should be included and agreement with literature data confirmed.
ANSWER: We are in possession of 31P data from a large cohort of normal volunteers. From these we extracted the data from age and gender matched controls for the glioblastoma patients. From these, the data of both supratentorial brain hemispheres was included to get representative data of the entire brain. The necessary changes can be found in all chapters throughout the manuscript.
- pH values from Pi chemical shift may be included
ANSWER: We calculated pH and Mg concentrations for the regions of interest, according to your suggested literature and provided the data in the results section. The results are very interesting, thank you very much for that recommendation! The necessary changes can be found in all chapters throughout the manuscript.
Minor points
- Sentence in lines 146-148 is not clear
ANSWER: The sentence was unclear because a point was missing. The parted sentences should be comprehensibly now. (page 6, line 175)
Some printing errors: e.g.
- Line 84 “promotor” is ”promoter”
ANSWER: We changed the typing to “promoter”. (page 3, line 98)
- Line 124 “displaying-ATP” is “displaying g-ATP”
ANSWER: We changed the typing to “displaying γ-ATP”. (page 5, line 150)
Check commas and points
ANSWER: We thoroughly checked the entire manuscript with regard to punctuation. The necessary changes can be found in all chapters throughout the manuscript.
Reviewer 5 Report
The paper describes a pilot study on metabolic changes observed by 31P-MRS in a short cohort of GBM patients. The main result is relative to the observation that changes are observed in different region of the brain, even in distant brain region.
Some limitations of the study are indicated by the authors. My main criticisms is relative to the absence of metabolic data on brain of healthy volunteers. As authors claim that metabolic changes are observed even in the opposite brain hemisphere, reference values for observed parameters in the normal brain under same experimental conditions should be presented. This point should be addressed.
A table to compare parameter ratios here presented with those of the literature for different tumor/brain regions could help the discussion on the meaning of changes of different energy related ratios and markers of membrane turnover .
Captions of Figures 1 and 2 are exchanged.
Author Response
The paper describes a pilot study on metabolic changes observed by 31P-MRS in a short cohort of GBM patients. The main result is relative to the observation that changes are observed in different region of the brain, even in distant brain region.
- Some limitations of the study are indicated by the authors. My main criticisms is relative to the absence of metabolic data on brain of healthy volunteers. As authors claim that metabolic changes are observed even in the opposite brain hemisphere, reference values for observed parameters in the normal brain under same experimental conditions should be presented. This point should be addressed.
ANSWER: We are in possession of 31P data from a large cohort of normal volunteers. From these we extracted the data from age and gender matched controls for the glioblastoma patients. From these, the data of both supratentorial brain hemispheres was included to get representative data of the entire brain. The necessary changes can be found in all chapters throughout the manuscript.
- A table to compare parameter ratios here presented with those of the literature for different tumor/brain regions could help the discussion on the meaning of changes of different energy related ratios and markers of membrane turnover.
ANSWER: We added a new table (table 2), which contains the information of seven prior studies on 31P MRS in patients with GBM. Unfortunately, these studies only compared tumor regions with either the contralateral hemisphere or the brains of normal controls. The present study is the first one comparing other regions with this method. However, our results are in excellent accordance with these prior studies. We tried to visualize that with ↑ and ↓ arrows to show in which area (tumor or control) the respective value was higher or lower than in the respective other area. (page 11, line 302)
- Captions of Figures 1 and 2 are exchanged.
ANSWER: We changed the figure captions for figure 1 and figure 2 to their correct positions. (page 4, lines 116 – 120; page 5, lines 154 – 156)
Round 2
Reviewer 2 Report
Please find the comments in the attached file

Author Response
Thank you very much for your comments! You can find our answers in the file attached.

Reviewer 4 Report
The manuscript has been revised according to the requests and missing data have been included.
After a careful analysis of the revised version I think that the manuscript is now scientifically sound and deserves publication.
Author Response
Thank you very much for your comments and for the recommendation of publication!
Reviewer 5 Report
The introduced modifications meet my requests and now I recommend publication of the paper.
Author Response

(The authors gave the same response as above.)
